# Hiding Images in Deep Probabilistic Models

**Haoyu Chen**
Department of Computer Science
City University of Hong Kong
haoychen3-c@my.cityu.edu.hk

**Linqi Song**
Department of Computer Science
City University of Hong Kong
linqi.song@cityu.edu.hk

**Zhenxing Qian**
School of Computer Science
Fudan University
zxqian@fudan.edu.cn

**Xinpeng Zhang**
School of Computer Science
Fudan University
zhangxinpeng@fudan.edu.cn

**Kede Ma**[*]
Department of Computer Science
City University of Hong Kong
kede.ma@cityu.edu.hk

## Abstract

Data hiding with deep neural networks (DNNs) has experienced impressive successes in recent years. A prevailing scheme is to train an autoencoder, consisting of an *encoding* network to embed (or transform) secret messages in (or into) a carrier, and a *decoding* network to extract the hidden messages. This scheme may suffer from several limitations regarding practicability, security, and embedding capacity. In this work, we describe a different computational framework to hide images in deep probabilistic models. Specifically, we use a DNN to model the probability density of cover images, and hide a secret image in one particular location of the learned distribution. As an instantiation, we adopt a SinGAN, a pyramid of generative adversarial networks (GANs), to learn the patch distribution of one cover image. We hide the secret image by fitting a deterministic mapping from a fixed set of noise maps (generated by an embedding key) to the secret image during patch distribution learning. The stego SinGAN, behaving as the original SinGAN, is publicly communicated; only the receiver with the embedding key is able to extract the secret image. We demonstrate the feasibility of our SinGAN approach in terms of extraction accuracy and model security. Moreover, we show the flexibility of the proposed method in terms of hiding multiple images for different receivers and obfuscating the secret image.

## 1   Introduction

Data hiding generally refers to the process of hiding a form of secret message in another form of cover media, while minimizing the introduced distortions to the cover media [14, 47]. For human eavesdroppers, the measured distortion should be consistent with human judgments, penalizing errors that are most perceptually or cognitively noticeable [55, 5]; for machine eavesdroppers, the distortion should be "invisible" in a way that bypasses digital steganalysis tools such as StegExpose [9] and more recent deep learning-based ones [10]. Only the informed receiver typically with a shared embedding key (through a secure subliminal channel [50]) is able to extract the secret message. The

---

[*]Corresponding author.

36th Conference on Neural Information Processing Systems (NeurIPS 2022).

form of the secret message can be encrypted bit streams [15], texts [30], audio signals [37], images [6], and videos[52]. Similarly, the cover media can also be texts [8], audio signals [31], images [6], videos [52], neural networks [5, 56], and even human behaviours[66].

Like many problems in signal and image processing, data hiding has been revolutionized by the remarkable development of DNNs [6, 67, 28]. These methods typically follow an autoencoder approach with two key components: an *encoding* network and a *decoding* network. For *secret-in-image* hiding [6, 67], the encoding network takes the cover image and the secret message as inputs, and generates a stego image with the hidden message (see Fig. 1 (a)). For *secret-in-network* hiding [5, 56], the cover media becomes some pre-selected weight layers of the encoding network, and the secret message usually serves as a watermark (see Fig. 1 (b)). For *constructive* (or generative) image hiding [28], the encoding network directly maps the secret message to the stego image without reliance on any cover image (see Fig. 1 (c)). In all cases, the decoding network is responsible for extracting the secret message. Despite demonstrated success, the autoencoder scheme may suffer from three main drawbacks. First, the decoding network, whose size may be significantly larger than that of the secret message, must be sent to the receiver side via the subliminal channel[2], making the paradigm less practical. Second, it is not hard to re-train existing DNN-based steganalysis methods [10, 57] to identify stego images (or stego weight matrices), making the paradigm less secure. Third, it is difficult to hide multiple images for different receivers via the same encoding and decoding networks, making the paradigm less flexible.

**Our Contributions**. In this paper, we propose to hide images in deep probabilistic models, which is substantially different from the previous autoencoder scheme (see Fig. 1 (d)). The key idea is to use a DNN to model the high-dimensional probability density of training cover images, and hide the secret image in one particular location of the learned distribution. The stego DNN for density estimation is publicly communicated, from which we may draw samples that look like training cover images. Only guided sampling by the embedding key (shared between the sender and receiver) is able to reproduce the secret image.

We construct a specific example under the proposed probabilistic image hiding framework. Specifically, we adopt a SinGAN [49], a pyramid of generative adversarial networks (GANs) [21], to implicitly learn the patch distribution of a single cover image. During distribution learning, we use the same SinGAN to fit a deterministic mapping from a fixed set of noise maps (generated by the shared embedding key) to the secret image, which completes the image hiding process. The stego SinGAN that behaves like the original one is publicly communicated. A single forward propagation suffices to extract the secret image by the receiver with the embedding key, and no decoding network is trained and transmitted. Experiments demonstrate that the proposed method is 1) feasible, extracting the secret image with improved accuracy (compared to the autoencoder-based methods), 2) secure, behaving normally as the original SinGAN in several aspects, and 3) flexible, hiding multiple images for different receivers and obfuscating the secret image with graceful performance degradation.

## 2   Related Work

Depending on the applications, data hiding can be broadly divided into two subtopics: watermarking and steganography. Watermarking [24] aims to embed a digital watermark in a multimedia file for copyright protection and content management on social networks. Thus, watermarking techniques focus primarily on the robustness and perceptibility aspects of the embedded watermarks. Steganography [33, 47] aims to conceal a secret message within a cover media mainly for covert communication. Thus, steganography pays more attention to the trade-off among embedding capacity, extraction accuracy, and model security. Here we provide a concise review of image hiding techniques, and refer the interested readers to [47, 14, 63] for more comprehensive surveys of the field.

**Secret-in-Image Hiding**. The most common image steganography modifies the least significant bits (LSBs) of images, either uniformly [13, 42] or adaptively [46, 25], guided by the design of novel distortion functions. Other representative techniques include pixel value differencing [60], histogram shifting [45, 64], and recursive code construction [65]. Transform domain steganography [48, 27, 7]) have also been proposed with improved capacity and security. Often, these methods leave traces in the form of certain statistical irregularities, which can be easily revealed by simple

---

[2]The decoding network, if shared via a public channel, will be suspectable, as it is only trained to extract the secret message rather than performing typical machine learning tasks.

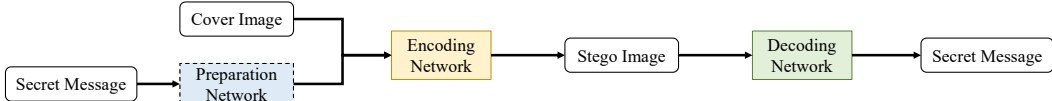

(a) Secret-in-image hiding. The preparation network is optional.

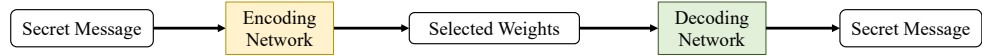

(b) Secret-in-network hiding. The secret message is usually in the form of a watermark.

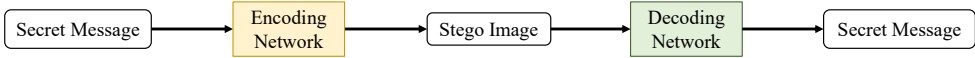

(c) Constructive (or generative) image hiding.

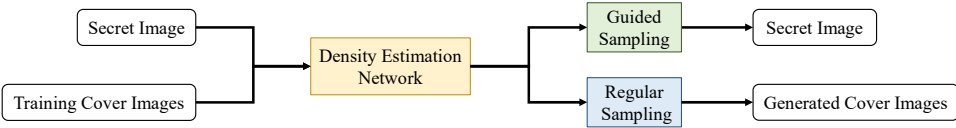

(d) Proposed framework for hiding images in deep probabilistic models.

Figure 1: Paradigms for hiding data using DNNs.

steganalysis algorithms as countermeasures [19, 41, 26]. Zhu *et al.* [67] proposed one of the first DNN-based autoencoder for unified watermarking and steganography with an optional noise layer. Baluja [6] extended it to "image-in-image" steganography with a preparation network to preprocess the secret image. Weng *et al.* [59] further extended it to "video-in-video" steganography via temporal residual modeling. Normalizing flow-based invertible architectures [18] have also been investigated for multiple image hiding [40, 32, 22]. Here, we describe a different framework to accomplish a similar but more challenging goal - multiple image hiding for different users - with several other advantages.

**Secret-in-Network Hiding**. A prerequisite for secret-in-network hiding is that the hidden message should not affect the network performance on the given machine learning task. As a result, the message (mostly the watermark for intellectual property protection of the neural network) is commonly embedded during network training. Typical strategies for this purpose include parameter regularization [53], backdooring [5], output watermarking [61], and weight selection [56]. The proposed framework can be seen as a form of secret-in-network hiding, but with different goals (steganography instead of watermarking).

**Constructive (Generative) Image Hiding**. Traditional methods hide secret messages during the construction of some specific types of images, such as textures [62] and fingerprints [36]. Recent DNN-based constructive image hiding methods mainly aim to construct the mapping between secret messages and stego images of more unconstrained content types [38, 58]. The proposed framework can be seen as a form of constructive image hiding, where we hide a secret image during the "construction" of a probability density function, but with larger embedding capacity and improved model security.

## 3 Hiding Images in Deep Probabilistic Models

**General Framework**. Without loss of generality, we describe the general framework of hiding a single image in deep probabilistic models. The straightforward extension to multiple image hiding is described in Sec. 4.3. We assume a cover image dataset $\mathcal{D} = \{\boldsymbol{x}^{(1)}, \boldsymbol{x}^{(2)}, \ldots, \boldsymbol{x}^{(M)}\}$, where

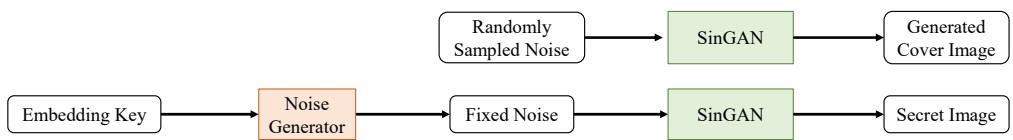

Figure 2: Hiding images in a SinGAN.

each image is drawn independently from an underlying image distribution $p(\boldsymbol{x})$. $\mathcal{D}$ may include a single image (where $\boldsymbol{x}^{(i)}$ becomes the $i$-th image patch), images of a specific class (*e.g.*, textures, fingerprints, and faces), and images on the natural image manifold. Also given is a secret image $\boldsymbol{x}^{(s)}$ of arbitrary content and an embedding key $\boldsymbol{k}$ shared by the sender and the receiver, which is a typical setting in the prisoner's problem formulated by Simmons [50].

The proposed probabilistic image hiding framework consists of two main steps. First, we learn a probability density function $p_s(\boldsymbol{x})$ over $\mathcal{D}_s = \mathcal{D}\bigcup\{\boldsymbol{x}^{(s)}\}$, *i.e.*, the combination of the cover image dataset and the secret image either explicitly via (approximate) maximum likelihood [35, 18, 51, 34] or implicitly via likelihood-free inference (*e.g.*, density estimation by comparison) [43]. The learned probability model $p_s(\boldsymbol{x})$ is publicly communicated in the proposed framework, which is usually in the form of a DNN (as the scoring function [20] or the sample generating function [43]). We may as well learn a *reference* probability density function $p_c(\boldsymbol{x})$ on the cover image dataset $\mathcal{D}$ solely. To guarantee the security of our framework, $p_s(\boldsymbol{x})$ should be as close to $p_c(\boldsymbol{x})$ as possible in some statistical distance (*i.e.*, distribution-preserving [12]). Provided that the adopted probability density estimator is robust (to the outlier image $\boldsymbol{x}^{(s)}$), which assigns an infinitesimal probability mass to $\boldsymbol{x}^{(s)}$, the proposed framework is perfectly secure in any statistical distance sense. Second, we design a guided sampling procedure (with the help of the embedding key $\boldsymbol{k}$) to draw a sample image $\hat{\boldsymbol{x}}^{(s)}$ from $p_s(\boldsymbol{x})$ that looks identical to the secret image $\boldsymbol{x}^{(s)}$. A third party without the embedding key is only able to generate samples that resemble cover images. It is noteworthy that the proposed framework does not require the learned $p_s(\boldsymbol{x})$ (or equivalently $p_c(\boldsymbol{x})$) to be sufficiently close to the underlying $p(\boldsymbol{x})$, a daunting (if not impossible) task to complete as digital images reside in a very high-dimensional space. In other words, it is perfectly fine that the samples drawn from $p_s(\boldsymbol{x})$ (without guidance) contain visually noticeable distortions as long as such distortions are shared by the samples drawn from $p_c(\boldsymbol{x})$.

**Specific Example**. Within the general framework of probabilistic image hiding, we provide a specific example that relies on GANs [21], a family of *implicit* probability models, which are represented by a stochastic procedure of data sampling. Although more natural to work with in our context compared to *prescribed* probabilistic models [16], unconditional GANs have the notorious reputation of being difficult to train, especially when modeling image sets with diverse content complexities and rich semantics. Thus, to make our work easily reproducible, we opt for a SinGAN [49] to learn the internal patch distribution of a single cover image $\boldsymbol{x}_0^{(c)}$ at various scales.

A SinGAN consists of a pyramid of generators $\{\boldsymbol{G}_0, \boldsymbol{G}_1 \ldots, \boldsymbol{G}_N\}$, where $\boldsymbol{G}_n$ takes the upsampled version of the generated image by $\boldsymbol{G}_{n+1}$ as well as an additive white Gaussian noise map $\boldsymbol{z}_n$ of the same size as inputs, and produces an image sample at the $n$-th scale:

$$\hat{\boldsymbol{x}}_n^{(c)} = \boldsymbol{G}_n\left(\boldsymbol{z}_n, \left(\hat{\boldsymbol{x}}_{n+1}^{(c)}\right)\uparrow^r\right), \quad n < N, \tag{1}$$

where $r > 1$ is the pre-defined upsampling ratio. The generation process starts at the coarsest scale (*i.e.*, the $N$-th scale), where the input is purely noise:

$$\hat{\boldsymbol{x}}_N^{(c)} = \boldsymbol{G}_N\left(\boldsymbol{z}_N\right), \tag{2}$$

and progressively makes use of all generators to produce the finest scale image $\hat{\boldsymbol{x}}_0^{(c)}$ with possibly different size and aspect ratio of $\boldsymbol{x}_0^{(c)}$ [49]. Coupled with the generators is a pyramid of discriminators $\{D_0, D_1, \ldots, D_N\}$, and each $D_n$ is trained to discriminate between patches extracted from $\hat{\boldsymbol{x}}_n^{(c)}$ (in Eq. (1)) and $\boldsymbol{x}_n^{(c)}$, which is a downsampled version of $\boldsymbol{x}_0^{(c)}$ by a factor of $r^n$ (*i.e.*, downsampled $n$ times by a factor of $r$).

Table 1: Extraction accuracy comparison when hiding one image. "↑": larger is better, and vice versa.

| Method | #params | PSNR↑ | SSIM↑ | DISTS↓ |
|--------|---------|-------|-------|--------|
| LSB | — | 23.06 | 0.785 | 0.095 |
| Baluja17 | 0.48M | 25.91 | 0.874 | 0.102 |
| HiDDeN | 0.38M | 27.99 | 0.897 | 0.096 |
| Weng19 | 42.6M | 35.64 | 0.942 | 0.055 |
| HiNet | 4.05M | 35.59 | 0.952 | 0.047 |
| Ours | 0.67M | **36.84** | **0.958** | **0.038** |

Table 2: Cover image quality, diversity, and weight distribution similarity between the original and stego SinGANs in terms of SIFID, DS, and KLD.

| #images | SIFID↓ | DS↑ | KLD↓ |
|---------|--------|-----|------|
| 0 (ref) | 0.041 | 0.407 | 0 |
| 1 | 0.046 | 0.430 | 0.001 |
| 2 | 0.045 | 0.415 | 0.004 |
| 3 | 0.047 | 0.427 | 0.006 |
| 4 | 0.051 | 0.438 | 0.008 |

The training objective for each scale is a weighted combination of an adversarial term and a reconstruction term:

$$\min_{\boldsymbol{G}_n} \max_{D_n} \ell_{\mathrm{adv}}\left(\boldsymbol{G}_n, D_n; \boldsymbol{x}^{(c)}\right) + \lambda \ell_{\mathrm{rec}}\left(\boldsymbol{G}_n; \boldsymbol{x}^{(c)}\right), \tag{3}$$

where $\lambda$ is the trade-off parameter. The adversarial loss $\ell_{\mathrm{adv}}$ is for penalizing the statistical difference between the patch distribution of the generated $\hat{\boldsymbol{x}}_n^{(c)}$ and that of the (downsampled) cover image $\boldsymbol{x}_n^{(c)}$. The reconstruction loss $\ell_{\mathrm{rec}}$ is for stabilizing the training process by ensuring that $\boldsymbol{x}_n^{(c)}$ can be reconstructed from a specific set of input noise maps. Following the original SinGAN paper [49], we use the WGAN-GP loss [23] and the mean squared error (MSE) to implement $\ell_{\mathrm{adv}}$ and $\ell_{\mathrm{rec}}$, respectively.

After training, the SinGAN is capable of generating new image samples that preserve the patch distribution of $\boldsymbol{x}^{(c)}$, with novel and plausible scene configurations and structures. Once the learning procedure of the SinGAN is clear, hiding the secret image $\boldsymbol{x}^{(s)}$ during the patch distribution learning can be straightforwardly done by modifying the training objective from Eq. (3) to

$$\min_{\boldsymbol{G}_n} \max_{D_n} \ell_{\mathrm{adv}}\left(\boldsymbol{G}_n, D_n; \boldsymbol{x}^{(c)}\right) + \lambda \ell_{\mathrm{rec}}\left(\boldsymbol{G}_n; \boldsymbol{x}^{(s)}\right). \tag{4}$$

That is, a reconstruction loss is replaced to enforce that a specific set of input noise maps $\boldsymbol{z}^{(s)} = \{\boldsymbol{z}_0^{(s)}, \boldsymbol{z}_1^{(s)}, \ldots, \boldsymbol{z}_N^{(s)}\}$ is mapped to the secret image $\boldsymbol{x}^{(s)}$ instead of the cover image $\boldsymbol{x}^{(c)}$. $\boldsymbol{z}^{(s)}$ can be generated by a standard Gaussian pseudo-random number generator [11] using the embedding key $\boldsymbol{k}$ as the seed. We may as well put back the reconstruction term for $\boldsymbol{x}^{(c)}$, but we find this makes little difference during training and testing. We conjecture that the reconstruction loss in Eq. (4) not only enables hiding of the secret image, but also plays a similar role in improving the training stability and convergence. For ease of description, we refer to models optimized for Eq. (3) and Eq. (4) as the *original* and *stego* SinGANs, respectively.

To extract the secret image $\boldsymbol{x}^{(s)}$, the receiver uses the shared embedding key $\boldsymbol{k}$ to re-generate the noise maps $\boldsymbol{z}^{(s)}$, which is fed to the publicly transmitted stego SinGAN for secret image extraction via a single forward propagation (see Fig. 2).

## 4 Experiments

In this section, we perform a series of experiments to verify the promise of our SinGAN approach. First, we evaluate secret image extraction accuracy both quantitatively and qualitatively in comparison to image-in-image hiding methods based on autoencoders. Second, we probe the security of the stego SinGAN by comparing it to the original one in terms of 1) quality and diversity of generated cover images, 2) marginal distribution similarity of model parameters [56], and 3) possibility of secret image leakage. Third, we experiment with our method in two more challenging scenarios: 1) hiding multiple images within a SinGAN for different users and 2) hiding the content-obfuscated image. We implement the generators and discriminators of the SinGAN by medium-size DNNs, whose specifications and training details are given in the Appendix. Our quantitative experiments make use of 200 test image pairs, whose details are also given in the Appendix.

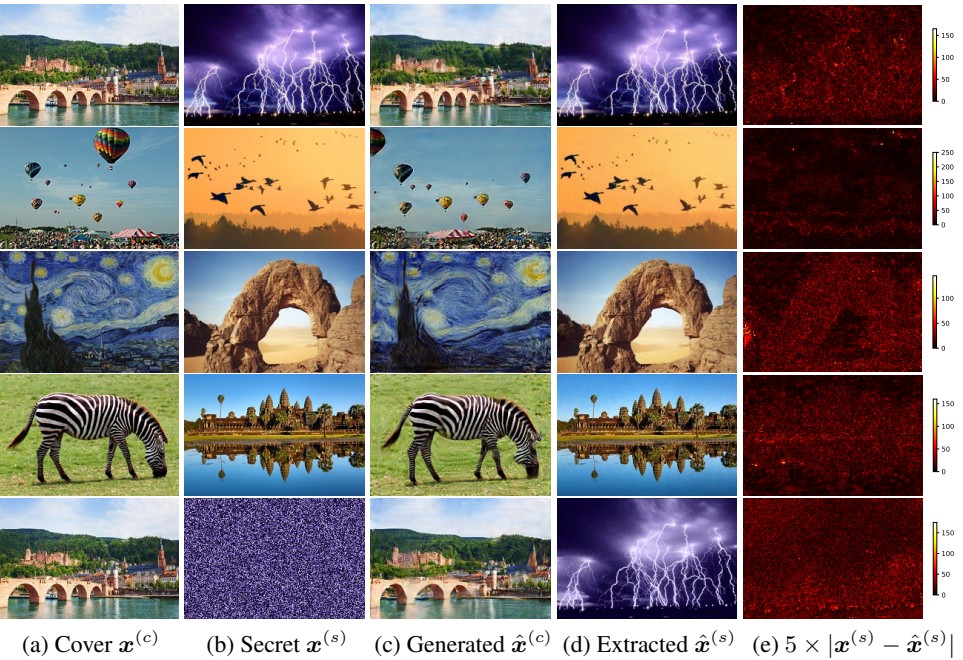

(a) Cover $\boldsymbol{x}^{(c)}$    (b) Secret $\boldsymbol{x}^{(s)}$    (c) Generated $\hat{\boldsymbol{x}}^{(c)}$   (d) Extracted $\hat{\boldsymbol{x}}^{(s)}$   (e) $5 \times |\boldsymbol{x}^{(s)} - \hat{\boldsymbol{x}}^{(s)}|$

Figure 3: Visual results of our SinGAN approach. The second image of the last row is a pixel-shuffled version of the second image in the first row. Zoom in for improved visibility.

## 4.1 Extraction Accuracy

In image data hiding, extraction accuracy means how the secret image can be faithfully reproduced at the receiver side. Here, we apply three objective image quality measures to quantitatively evaluate the signal fidelity, perceptual fidelity, and perceptual quality of the extracted secret image: the peak signal-to-noise ratio (PSNR), the structural similarity (SSIM) index [55], and the deep image structure and texture similarity (DISTS) measure [17]. Although our SinGAN approach is the first of its kind in the proposed probabilistic image hiding framework, we compare it with one naïve LSB replacement method, and four image-in-image steganography methods - Baluja17 [6], HiDDeN [67], Weng19 [59] and HiNet[32]. The LSB replacement method simply replaces the four LSB planes of the cover image with the four most significant bit (MSB) planes of the secret image. As the robustness is not our focus, we re-train HiDDeN [67] for image hiding without the noise layer. Weng19 [59] is a DNN-based video steganography method, and only the image hiding branch is used for testing. HiNet [32] relies on a normalizing flow-based invertible DNN to implement the encoder, such that the decoder is simply its inverse. We use the publicly available implementations for Baluja17 [1], HiDDeN[2], Weng19[3] and HiNet[4] with default training and testing[3] settings, and implement the LSB replacement by ourselves.

The extraction accuracy results are shown in Table 1, where we find that the proposed SinGAN approach performs favorably against existing autoencoder-based image-in-image steganography methods. We consider the obtained improvements as significant because secret-in-network hiding is generally considered much more difficult than secret-in-image hiding, where the latter has significant spatial redundancy to be reduced. After all, the most recent secret-in-network hiding method [56] has a limited embedding capacity up to $6,000$ bits. We also show some representative visual results of our method in Fig. 3. It is clear that the stego SinGAN is able to capture the internal patch distribution of the cover image, generating image samples with different but reasonable structures and configurations

---

[3]It is important to note that the results of autoencoder-based methods in our paper (especially in terms of extraction accuracy in Table 1) may be noticeably different from some of the previous publications. This is because we choose to quantize the stego image from the single-precision floating-point format of $32 \times 3$ to $8 \times 3$ bits per pixel before transmitting it to the receiver side. If such quantization is not properly enforced, trivial hiding solutions may exist because there are just more space to accommodate the cover and secret images by a simple concatenation.

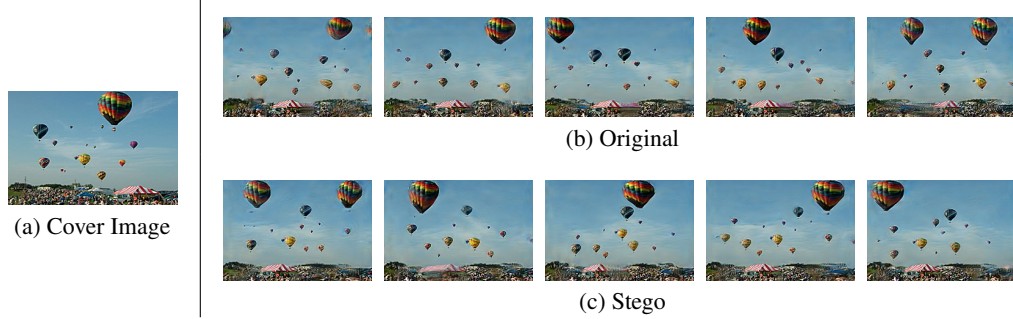

(a) Cover Image

(b) Original

(c) Stego

Figure 4: Visual comparison of the generated cover images "Fire balloons" by the original and stego SinGANs. In our case, the secret is the "Birds" image, as shown in the second row of Fig. 3.

of the same natural scene. Moreover, the extracted secret image by guided sampling is visually close to the original, which is adequate for use in the majority of real-world steganography applications.

## 4.2 Model Security

Steganographic analysis (*i.e.*, steganalysis) aims to identify whether there are hidden messages in suspected media, which constitutes an indispensable part of steganography evaluation. Traditional statistics-based and recent DNN-based steganalysis methods [9, 10] are mostly designed to discriminate between cover and stego images, thus not applicable when the cover media is a DNN. To the best of our knowledge, we are not aware of a steganalysis algorithm that accepts a full DNN (with millions of parameters) as input. Instead, we propose to probe the security of our SinGAN approach from the following three different aspects.

**Quality and Diversity of Generated Cover Images**. In the proposed framework, the stego SinGAN is publicly transmitted, and thus it must function as the original SinGAN [49]. As the primary goal of SinGANs is to model internal patch distributions, we examine and compare the quality and diversity of generated cover images by the original and stego SinGANs. To quantify quality, we adopt the single image Fréchet inception distance (SIFID) metric, as suggested in [49]. To quantify diversity, for each cover image, we compute the diversity score (DS) as the standard deviation (std) of each pixel values over 25 generated samples of the cover content, averaged over all pixels and normalized by the std of pixel intensities of the cover image [49]. The average quality and diversity results over 200 test image pairs are shown in Table 2, where we find that both SIFID and DS of the stego SinGAN are statistically indistinguishable from those of the original SinGAN based on a hypothesis testing using $t$-statistics [44]. Fig. 4 shows some randomly sampled examples from the original and stego SinGANs, which provides additional visual evidence that they learn very similar internal patch distributions of the cover image.

**Marginal Distribution Similarity of Model Weights**. As the proposed method essentially hides the secret image in the learned weights of the stego SinGAN, it is natural to ask whether its weight distributions significantly deviate from those of the original SinGAN. Following [56], we compute the Kullback–Leibler divergence (KLD) between the marginal distributions of model parameters of the original and stego SinGANs, as shown in the last column of Table 2. We find that the marginal distributions of the two sets of model parameters are almost identical, as evidenced by a KLD close to zero. Similar results can be obtained if the marginal distributions are compared in a per-stage fashion (see more results in the Appendix with visual comparison of histograms).

**Possibility of Secret Image Leakage**. One may also wonder the possibility of secret image leakage if an adversary constantly draws samples from the stego SinGAN. Such steganalysis arises naturally from the fact that there is no theoretical guarantee that the mapping between the secret noise (generated by the embedding key) and the secret image is bijective (*i.e.*, one-to-one). In other words, there might be some other sets of noise that are also mapped to the secret image, or at least some of its semantically meaningful content. Since the SinGAN is trained to be a black-box sampler, it is difficult to provide a theoretical analysis of the possibility of secret image leakage. Nevertheless, we conduct an empirical study, in which we randomly draw $100,000$ samples from each of the 200 trained stego SinGANs. Visual inspection of the thumbnails of generated samples indicates that no secret image

Table 3: Extraction accuracy of our method when hiding multiple images in one SinGAN.

| #images | PSNR↑ | SSIM↑ | DISTS↓ |
|---------|-------|-------|--------|
| One | 36.84 | 0.958 | 0.038 |
| Two | 35.91 | 0.946 | 0.043 |
| Three | 34.93 | 0.935 | 0.049 |
| Four | 34.03 | 0.923 | 0.055 |

Table 4: Extraction accuracy of our method with image obfuscation.

| Obfuscation | PSNR↑ | SSIM↑ | DISTS↓ |
|-------------|-------|-------|--------|
| No | 36.84 | 0.958 | 0.038 |
| Yes | 20.53 | 0.726 | 0.172 |

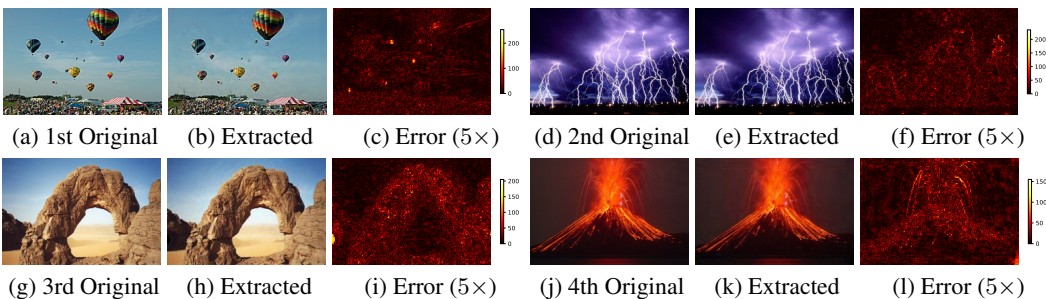

(a) 1st Original    (b) Extracted    (c) Error (5×)    (d) 2nd Original    (e) Extracted    (f) Error (5×)

(g) 3rd Original    (h) Extracted    (i) Error (5×)    (j) 4th Original    (k) Extracted    (l) Error (5×)

Figure 5: Visual comparison between the original and extracted secret images from a SinGAN with four hidden images.

(or images with similar semantics) is revealed. Therefore, it is safe to empirically conclude that the possibility of secret image leakage is less than $0.001\%$.

In summary, we have empirically proven that the proposed SinGAN approach is secure: hiding a full-size photographic image into a SinGAN does not compromise the quality and diversity of generated cover images, nor skew the weight distribution. It also survives constant sampling by an adversary. Such level of security verifies our claims that the secret image indeed occupies a tiny portion of probability mass of $p_s(\boldsymbol{x})$, and that $p_s(\boldsymbol{x})$ and $p_c(\boldsymbol{x})$ are statistically close.

### 4.3 Further Extensions

**Hiding Multiple Images for Different Receivers**. Hiding multiple images in a DNN is challenging; doing so for different receivers is even more challenging, and has not been accomplished before. Here the main difficulties lie not only in that more embedding capacity is required, but also in that each receiver must only extract her/his piece of message, and cannot extract (or even affirm the existence of) other messages [56]. The proposed probabilistic image hiding framework provides a straightforward and elegant extension to hide multiple images for different users, which follows a similar two-step approach. First, learn a probabilistic density function $p_s(\boldsymbol{x})$ over $\mathcal{D}_s = \mathcal{D} \bigcup \{\boldsymbol{x}^{(s_1)}, \ldots, \boldsymbol{x}^{(s_T)}\}$, where $T$ is the number of secret images and is considerably smaller than $M$, the number of cover images. Second, design $T$ guided sampling procedures using $T$ different embedding keys $\mathcal{K} = \{\boldsymbol{k}^{(1)}, \ldots, \boldsymbol{k}^{(T)}\}$, shared to $T$ different receivers. The learning goal remains the same: $p_s(\boldsymbol{x})$ should be close in some statistical distance to the reference distribution $p_c(\boldsymbol{x})$. The described procedure is more easily understood using the SinGAN instantiation, where we just modify the objective function in Eq. (4) to

$$\min_{\boldsymbol{G}_n} \max_{D_n} \ell_{\text{adv}}\left(\boldsymbol{G}_n, D_n; \boldsymbol{x}^{(c)}\right) + \lambda \frac{1}{T} \sum_{t=1}^{T} \ell_{\text{rec}}\left(\boldsymbol{G}_n; \boldsymbol{x}^{(s_t)}\right). \tag{5}$$

After training, the $t$-th receiver is able to re-generate the $t$-th specific set of noise maps $\boldsymbol{z}^{(s_t)}$ using the shared embedding key $\boldsymbol{k}^{(t)}$ for the $t$-th secret image extraction. S/he is, by design, ignorant of the presence (or absence) of other secret images. Even if the receiver is informed in some way that the current SinGAN contains multiple secret images, without extra embedding keys, s/he cannot extract images that are not intended to share with her/him.

We train SinGANs to hide up to four secret images, *i.e.*, $T \in \{2, 3, 4\}$. For each value of $T$, we train 200 SinGANs to hide different combinations of cover and secret images. The quality and diversity of the cover image, and the weight distribution similarity between the original and stego SinGANs are shown in Table 2, where we find that hiding multiple images does not seem to compromise the security of the proposed method. The extraction accuracy results and visual examples are shown in Table 3 and Fig. 5, respectively. With the number of increasing secret images, the extraction accuracy in terms of the three objective metrics degrades gracefully. The visual appearances of the extracted secret images are similar to those of the original, showing great promise of our method in hiding multiple images for different receivers.

**Obfuscating the Secret Image**. Inspired by [6], we consider obfuscating the secret image by shuffling its pixels (*i.e.*, image scrambling [54]), as a way of strengthening security. In this case, The shuffling key, together with the embedding key, is shared to the receiver. Table 4 shows the extraction accuracy results, where we see that pixel shuffling significantly increases the difficulty of image hiding. Nevertheless, from the last row of Fig. 3, we observe that the main content is clearly visible, despite somewhat noisy appearance.

## 5    Conclusion and Discussion

We have described a new computational framework for hiding images in deep probabilistic models, which is in stark contrast to previous steganography schemes. We provided an instantiation, where we used the SinGAN to build the internal patch distribution of the cover image, and hid the secret image during patch distribution learning. We conducted a series of experiments to demonstrate the feasibility of the proposed SinGAN approach in terms of extraction accuracy and model security. Moreover, our method is readily extended to hide multiple images for different receivers, a challenging task that has not been accomplished before. In addition, it works nicely with pixel shuffling, which adds additional security.

The current work opens the door to a new class of image hiding methods, with many interesting problems to be explored. First, the extraction accuracy of our SinGAN approach still has quite some room for improvement. For applications that require precise recovery of the secret image, it is worth exploring more efficient network structures, optimized for perceptual losses that exploit the physiological properties of the human visual system. Second, our method naïvely bypasses the steganalysis tools specifically designed for secret-in-image steganography. Currently, we have designed three different tests to probe the model security. As SinGANs [49] can be applied in a much wider range of image manipulation tasks such as super-resolution and paint-to-image translation, the stego SinGAN should be tested in those applications as well for model security. More importantly, we expect future effort to be dedicated to building steganalysis methods that accept a full DNN as input and assess whether it contains secret messages. Third, so far, we have just given the theoretical intuition of the probabilistic image hiding framework, supplied with empirical evidence. It is of mathematical interest to rigorously measure the statistical distance between $p_c(\boldsymbol{x})$ and $p_s(\boldsymbol{x})$, in an attempt to answer important questions like 1) where the secret image is hidden in the network (or equivalently the learned distribution) and 2) what the maximum number of secret images is allowed for a given distance constraint (*e.g.*, $\text{KLD}(p_c(\boldsymbol{x}) \parallel p_s(\boldsymbol{x})) \leq \epsilon$). Fourth, many other generative modeling methods [51] are worthy of deep investigation within the proposed framework to circumvent some of the limitations of the SinGAN approach. For example, we may model the (Stein) score function [39, 29] of $p_s(\boldsymbol{x})$ (*i.e.*, the gradient of $\log p_s(\boldsymbol{x})$), and develop guided Langevin-type sampling (by the embedding key) to extract the secret image.

## Acknowledgments and Disclosure of Funding

The authors would like to thank Song Yang for inspiring discussion. This work was supported in part by the Hong Kong RGC ECS Grants 21213821 (to KDM) and 21212419 (to LQS), the National Natural Science Foundation of China under Grants 62071407, 61936214, U20B2051 and U1936214, the InnoHK initiative, the Government of the HKSAR, Laboratory for AI-Powered Financial Technologies, and the Tencent AI Lab Rhino-Bird Gift Fund.

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
