# A Specifications and Training Details

## A.1 Model Architecture

The architecture of the SinGAN used in our paper follows that in [4]. As shown in Fig. 1, the generator at the $n$-th scale consists of a front-end convolution, $N - n + 1$ convolution blocks each with three convolution layers. Each convolution block, except for the last one, is followed by an upsampling layer with two residual connections, one for residual learning and one for incorporating a noise map from a higher scale (with a smaller spatial size). The last convolution block is responsible for the $n$-th scale image reconstruction with one residual connection. One back-end convolution with $\tanh()$ nonlinear activation is used to produce a scaled version of the RGB image as output. All convolution layers (except for the back-end one) have $64$ filters with a filter size of $3 \times 3$, followed by batch normalization and leaky ReLU activation with the negative slope of $0.05$. The weights of the front-end convolution, and the first $N - n$ convolution blocks, and the back-end convolution are inherited from those of the trained generator at the $n + 1$-th scale.

All generators share one discriminator, which is composed of five convolution layers all with a filter size $3 \times 3$. The first four convolution layers have $64$ filters followed by leaky ReLU activation with the negative slope of $0.05$, while the last convolution layer has a single convolution filter.

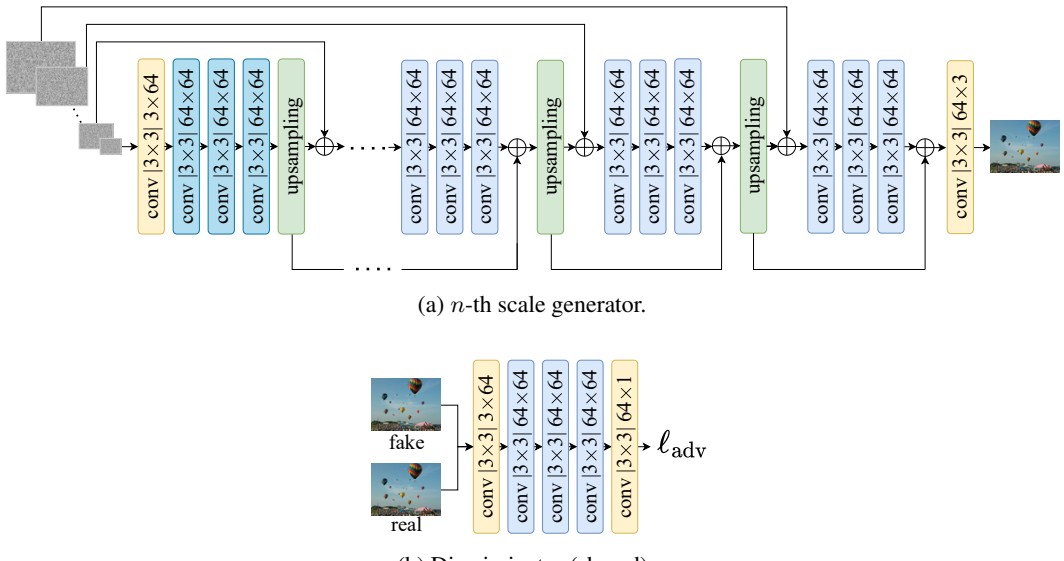

(a) $n$-th scale generator.

(b) Discriminator (shared).

Figure 1: Architecture of the SinGAN used in our paper.

## A.2 Optimization

We adopt the improved techniques for training SinGANs as recommended in [4]. The $\lambda$ in Eq. (4) is set to $10$. The trade-off parameter in WGAN-GP [3] is set to $0.1$ for gradient penalty. For the generator at the $n$-th scale, the newest three convolution blocks along with the front-end and back-end convolutions are jointly trained (or fine-tuned), while holding the older convolution blocks fixed (if any). This training strategy seems effective in preventing mode collapse. Adam[5] is adopted as the stochastic optimizer with an initial learning rate of $0.0005$ and a decay factor of $0.1$ after finishing $80\%$ of iterations, and we set the maximum number of training iterations to $2,000$. The training time is approximately 20 minutes for one image pair (each with size $244 \times 164 \times 3$) on an NVIDIA GeForce RTX3080 GPU.

# B Test Image Pairs for Quantitative Experiments

The 200 test image pairs used in our quantitative experiments are randomly drawn from five popular datasets (COCO[6], DIV2K[1], LSUN bedroom[7], ImageNet[2], Places[8]). Specifically, we sample

80 images from each dataset to obtain a total of 400 images, which are randomly partitioned into 200 cover images and 200 secret images, as shown in Figs. 2 and 3 respectively, with co-located images forming one pair.

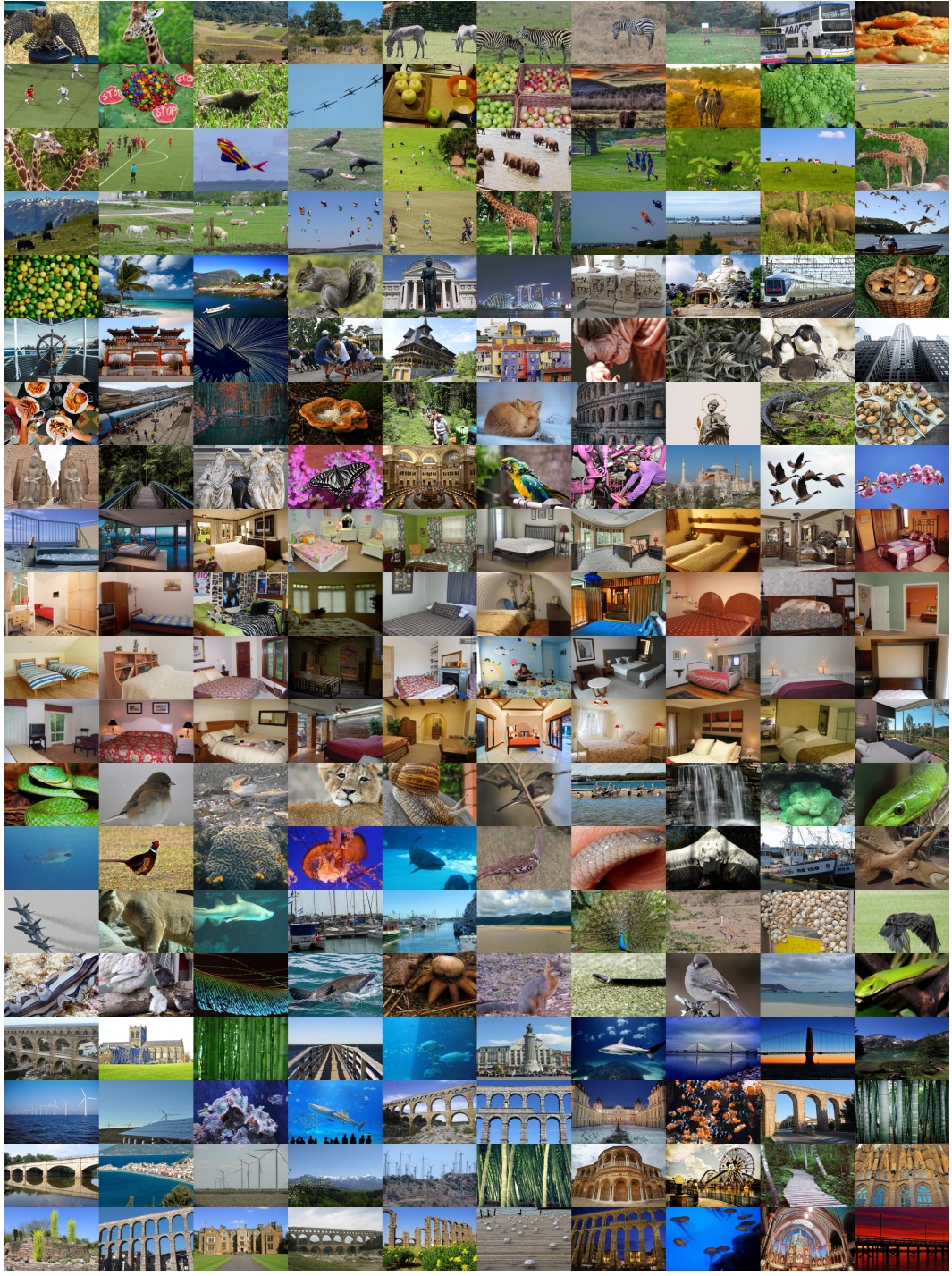

Figure 2: 200 cover images used for quantitative comparison.

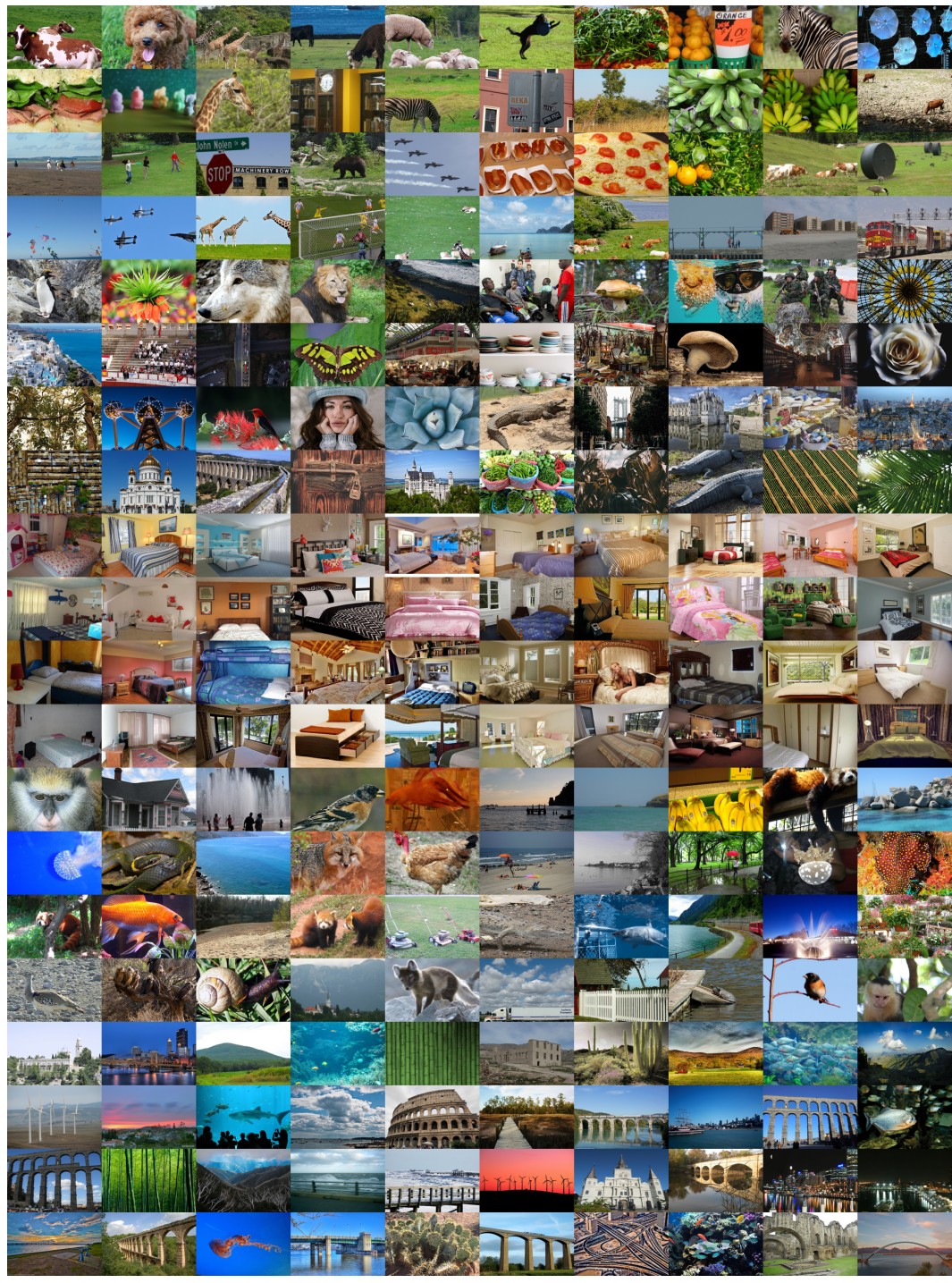

Figure 3: 200 secret images used for quantitative comparison. The two co-located cover and secret images form one pair.

## C  Histograms of Weight Distribution

### C.1  Total Weight Distribution

Fig 4 shows the histograms of all weights from 200 original and stego SinGAN generators. Careful visual inspection shows that the empirical weight distributions of the original and stego SinGAN generators are identical even when we hide up to four images.

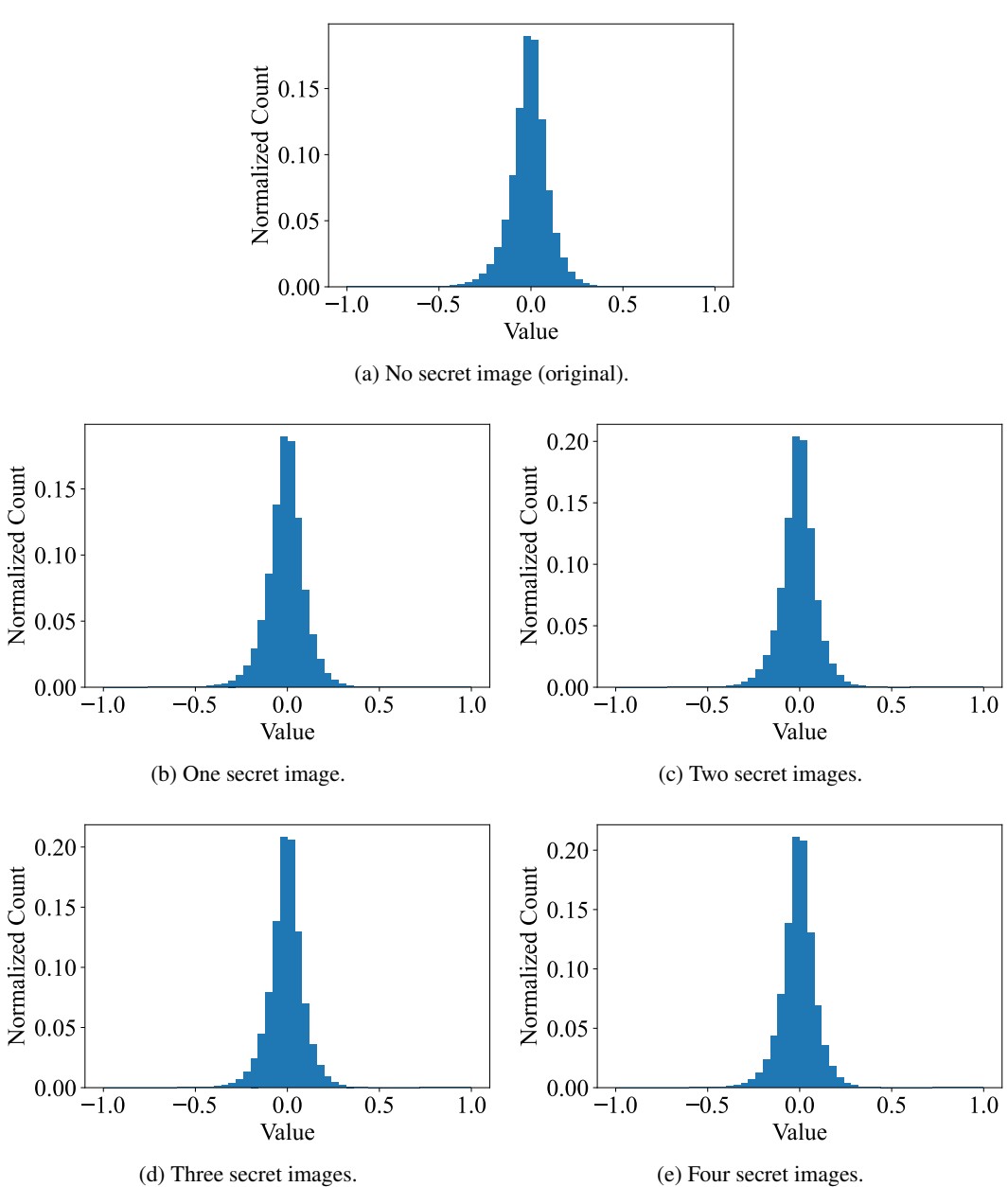

(a) No secret image (original).

(b) One secret image.

(c) Two secret images.

(d) Three secret images.

(e) Four secret images.

Figure 4: Visual comparison of histograms of the total weights.

### C.2  Per-Stage Weight Distribution

In addition to total weight distribution, the comparison of per-stage weight distribution is also provided. Here, one "stage" refers to a convolution block of the SinGAN generator, consisting of three convolution layers, as shown in Fig 1. Per-stage comparison gives us a finer view of the weight

distributions of the original and stego SinGAN generators. As our final generator has six stages (*i.e.*, six convolution blocks), we show six sets of per-stage weight distribution comparison results in Figs. 5 to 10. Here, the calculation of normalized count is with respect to one stage of weights across 200 SinGANs. Again, no visually noticeable differences can be observed.

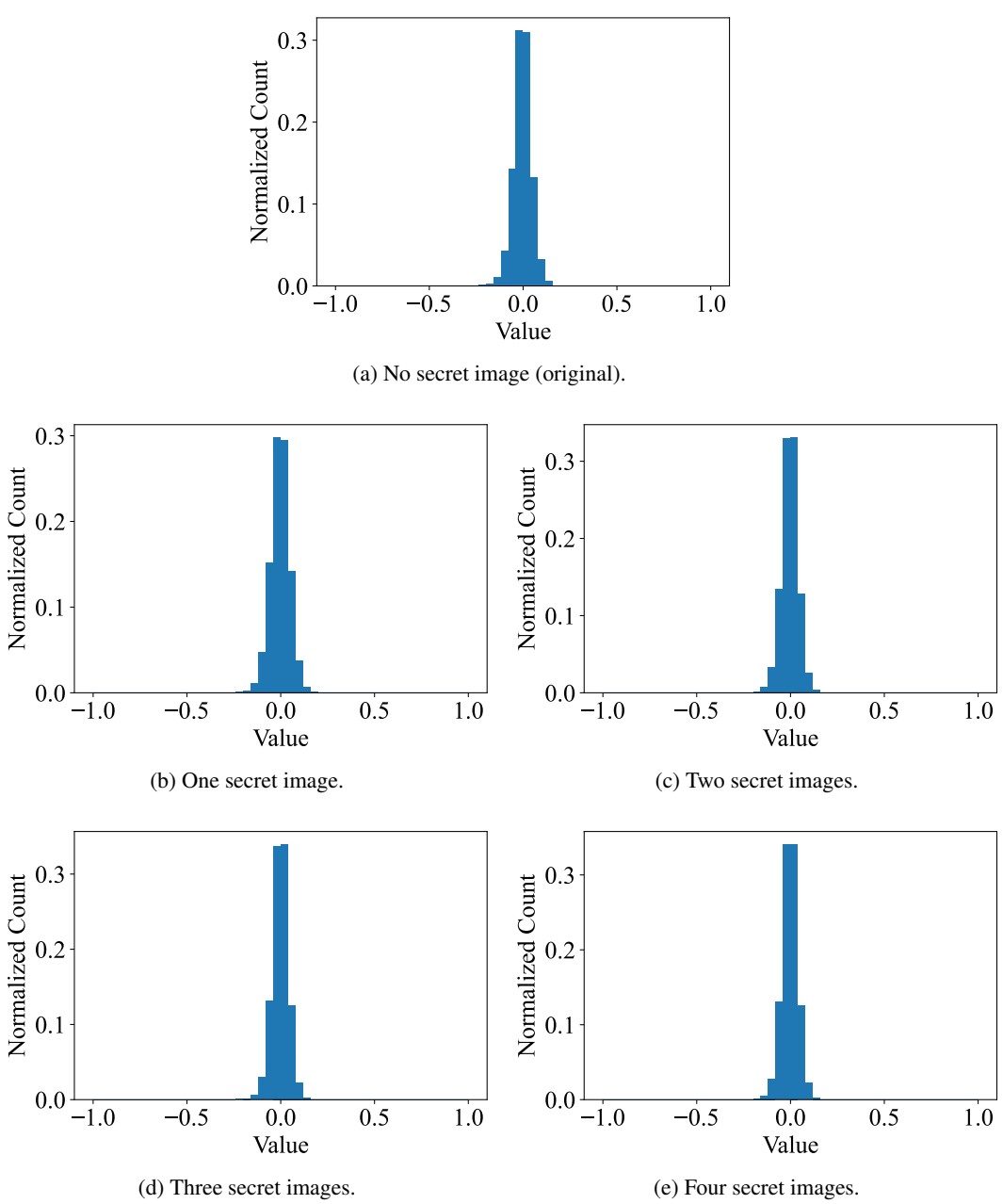

(a) No secret image (original).

(b) One secret image.

(c) Two secret images.

(d) Three secret images.

(e) Four secret images.

Figure 5: Visual comparison of histograms of the first-stage weights.

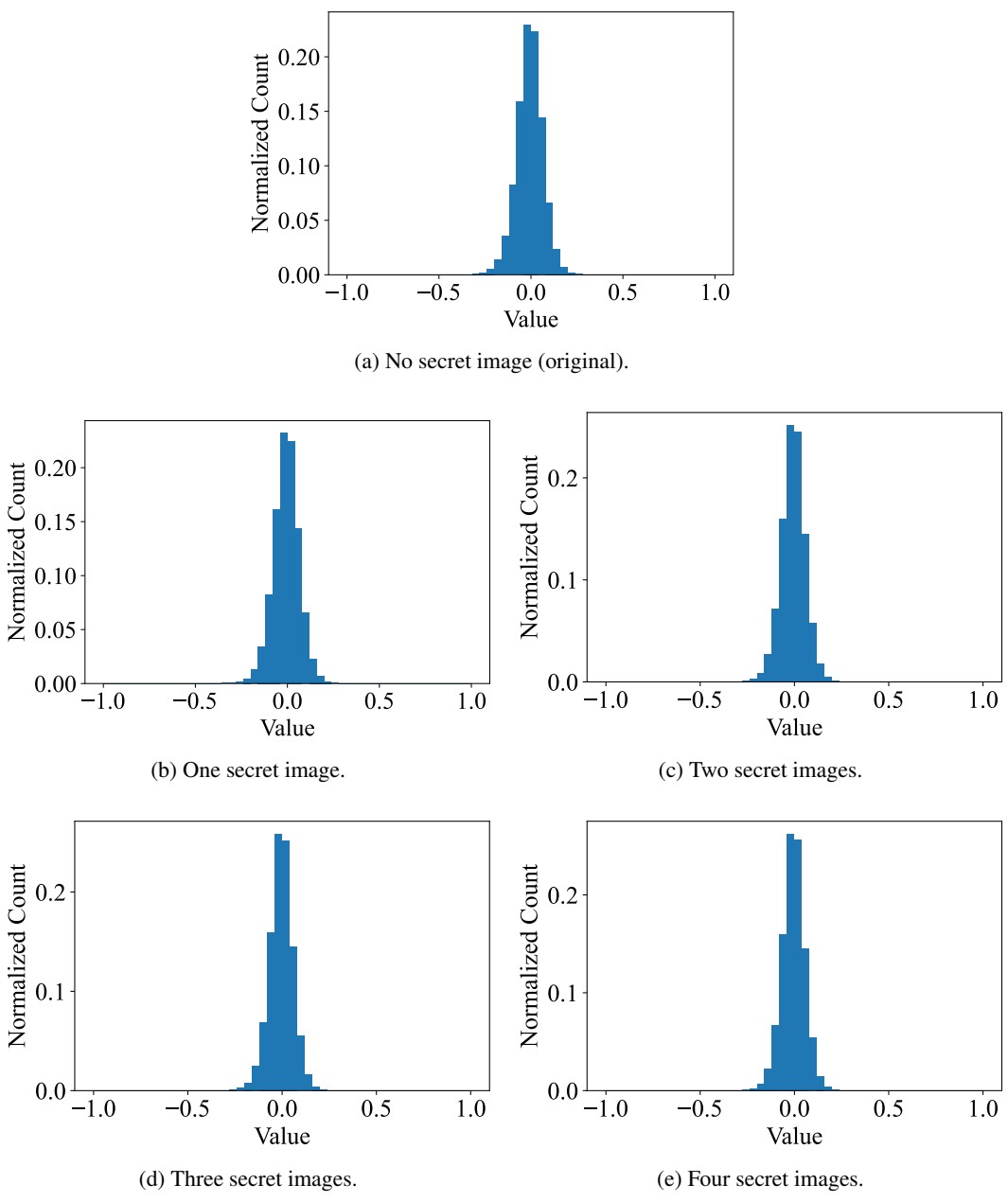

(a) No secret image (original).

(b) One secret image.

(c) Two secret images.

(d) Three secret images.

(e) Four secret images.

Figure 6: Visual comparison of histograms of the second-stage weights.

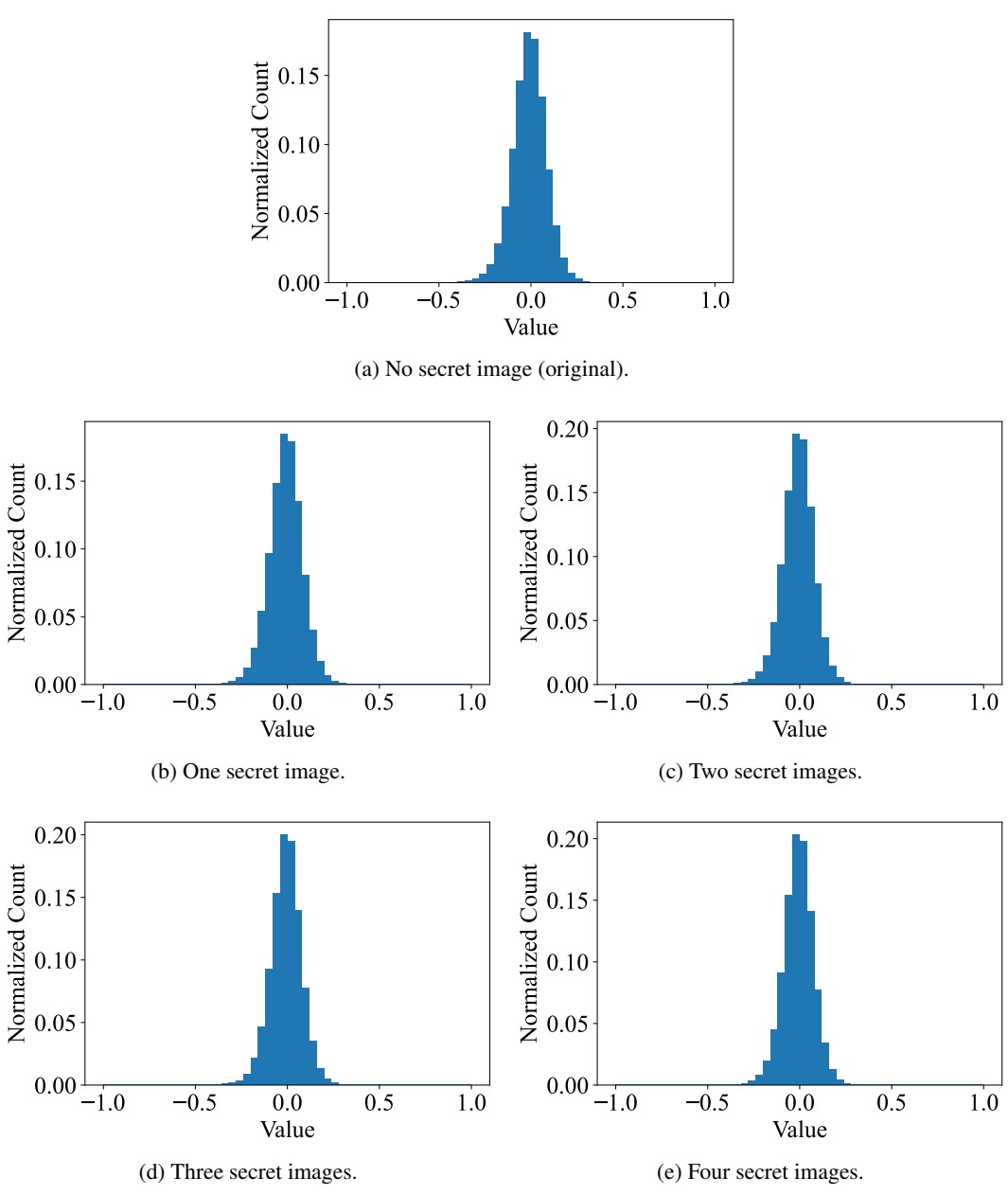

(a) No secret image (original).

(b) One secret image.

(c) Two secret images.

(d) Three secret images.

(e) Four secret images.

Figure 7: Visual comparison of histograms of the third-stage weights.

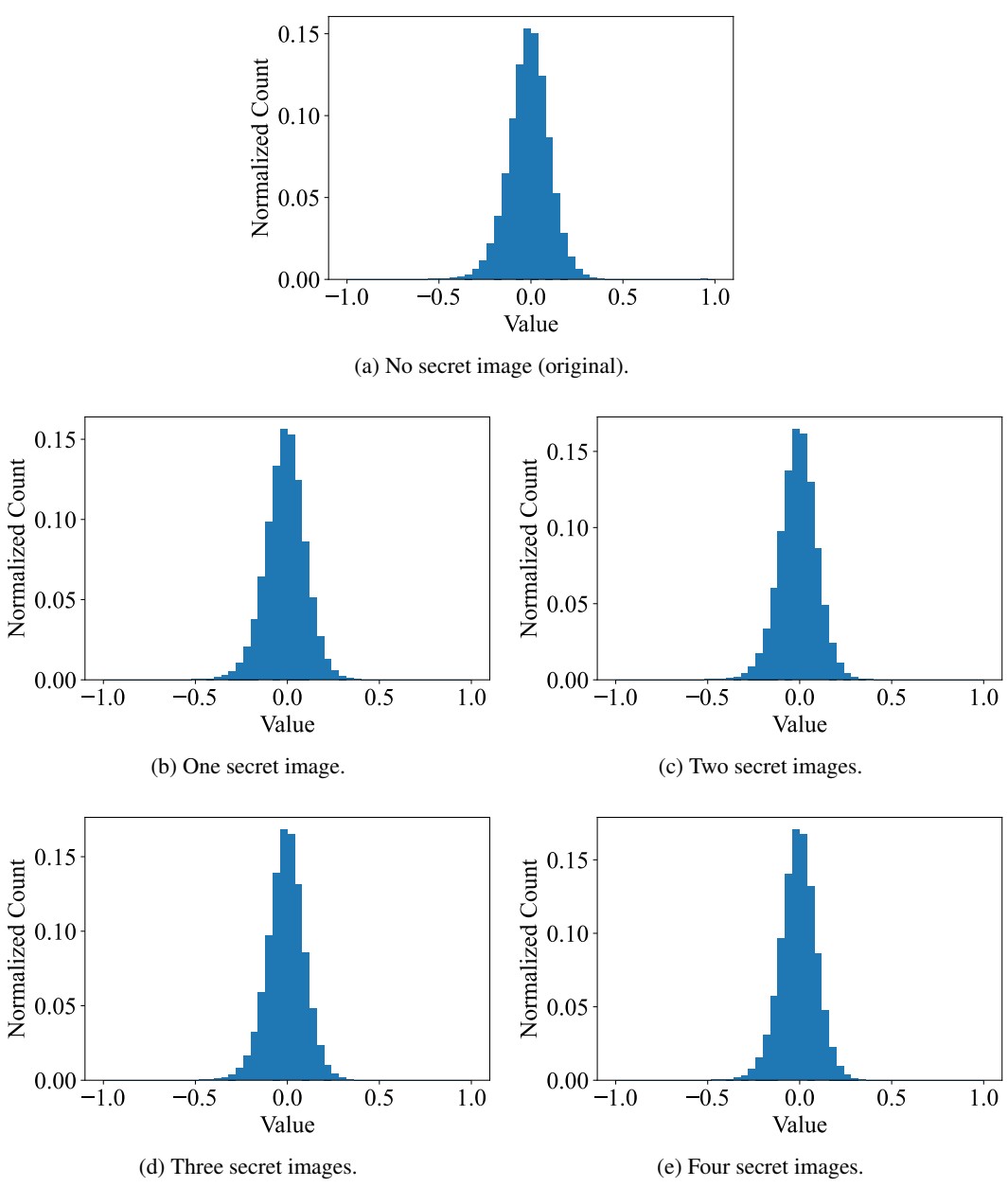

(a) No secret image (original).

(b) One secret image.

(c) Two secret images.

(d) Three secret images.

(e) Four secret images.

Figure 8: Visual comparison of histograms of the fourth-stage weights.

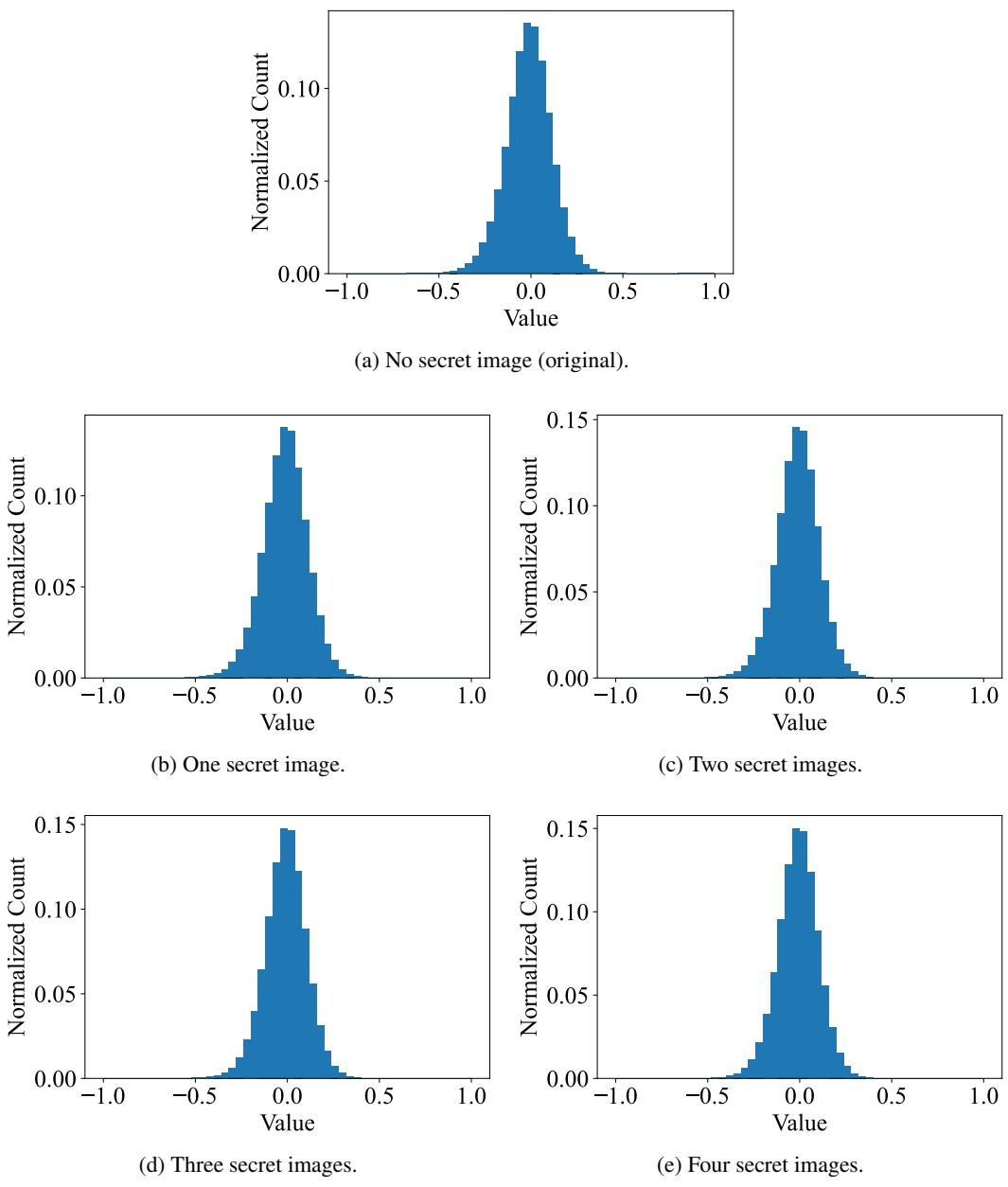

(a) No secret image (original).

(b) One secret image.

(c) Two secret images.

(d) Three secret images.

(e) Four secret images.

Figure 9: Visual comparison of histograms of the fifth-stage weights.

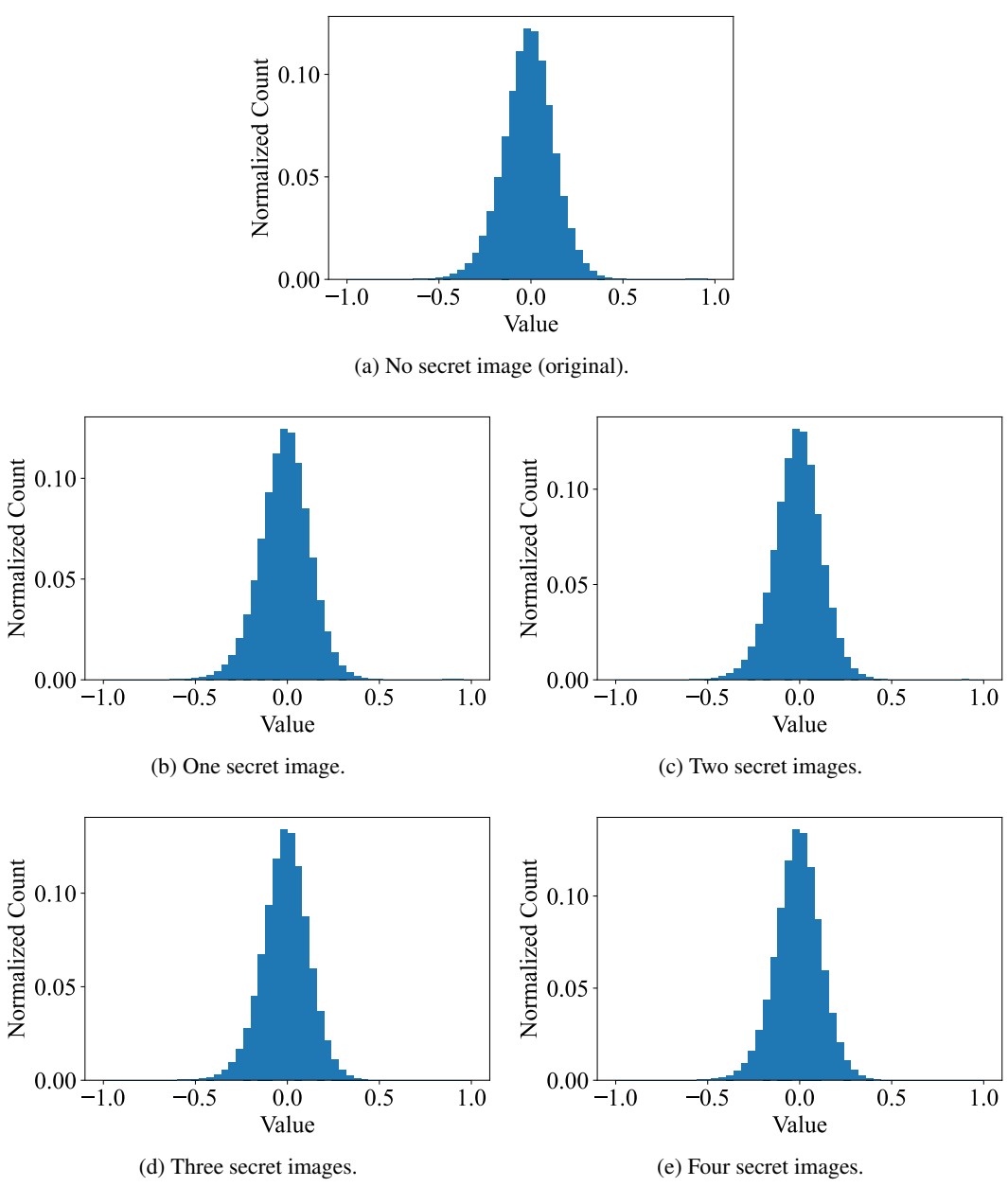

(a) No secret image (original).

(b) One secret image.

(c) Two secret images.

(d) Three secret images.

(e) Four secret images.

Figure 10: Visual comparison of histograms of the sixth-stage weights.