# OpenReview forum: "Hiding Images in Deep Probabilistic Models"
_NeurIPS.cc/2022/Conference — NeurIPS 2022 Accept_

### Official Review · Reviewer_bS3X · 2022-07-10

**Rating:** 7
**Confidence:** 3
**Soundness:** 3 good
**Presentation:** 3 good
**Contribution:** 3 good

**Summary:**

The paper proposes a method to hide images in deep probabilistic models. The proposed method is novel and can be framed as a particular case of the existing methods. One of the significant properties of the proposed method is its ability to be used for multiple receivers. The authors introduce the problem by discussing the current different approaches to hiding data using DNNs and outline the contributions of the paper. Next, related work for each of the described current approaches is provided. The authors then give a mathematical model formulation and how the data hiding computations are carried out. The approach being novel cannot be directly compared to previous works and hence, the authors identify the closest baselines to the proposed model and using experimental results, show that the proposed model either performs better or is more efficient than the considered baselines. The authors also evaluate the model from a security perspective using three methods. Lastly, the authors describe experiments evaluating the models' ability to hide images for multiple receivers and obfuscate the secret image.

**Questions:**

NA

**Strengths And Weaknesses:**

Typos:
1. Line 54: adopts -> adopt/adopted

---

> ### Author Response · Authors · 2022-07-29
> **Response to Reviewer bS3X**
>
> Thanks for appreciating our work. The summary of the current paper is indeed thorough and accurate.
>
> 1. We thank the reviewer to recognize the ability of the proposed method to hide multiple images for different receivers as a significant advantage over previous methods, despite that we choose to down-weight this part in our writing to give prominence to the new and general probabilistic hiding framework.
> 2. We appreciate the reviewer to recognize the primary contribution of the current work is the mathematical formulation of probabilistic image hiding, which is also computational feasible.

---

### Official Review · Reviewer_ShzF · 2022-07-13

**Rating:** 5
**Confidence:** 2
**Soundness:** 2 fair
**Presentation:** 2 fair
**Contribution:** 3 good

**Summary:**

The task is about image-in-image steganography. Unlike previous methods which follow the autoencoder approach, they use GAN-based network to model the probability density of cover images, and hide a secret image in one particular location of the learned distribution.

**Questions:**

(1) Lines 197-201 are unclear. The proposed method performs worse than Weng19 and HiNet in terms of the extraction accuracy of secret image. It is claimed that secret-in-network hiding is generally considered much more difficult than secret-in-image hiding. But at line 97, the proposed method is classified as constructive image hiding, not secret-in-network hiding. The authors should explain clearly why the extraction accuracy is low.

**Ethics Review Area:**

["I don’t know"]

**Limitations:**

The other methods can use one model to accommodate different cover-secret images. But for the proposed method, it needs to train one sinGAN for each cover image.

**Strengths And Weaknesses:**

Strengths
(1) The idea of hiding message in the learned distribution is interesting. It is different from previous methods, which use an autoencoder scheme.
(2) Compared with previous work, the proposed method doesn’t need to transmit the decoder to receiver via a subliminal channel. Their model can be publicly transmitted and only the key needs to be sent via the subliminal channel.
(3) The proposed method doesn’t directly generate stego images, thus avoiding the issue of possible detection by steganalysis methods.
(4) Multiple images can be hided in one model for different receivers, which is a challenging task that has not been accomplished before.

Weakness
(1) The dataset contains only 20 images.
(2) The other methods can use one model to accommodate different cover-secret images. But for the proposed method, it needs to train one sinGAN for each cover image.
(3) The proposed method performs much worse than HiNet in terms of the extraction accuracy of the secret image.
(4) The extraction accuracy of secret image with obfuscating has low PSNR (20dB).
(5) The mapping between the secret noise (generated by the embedding key) and the secret image may not be bijective (i.e., one-to-one), which means it is possible to obtain the secret image with random sample. In this paper they randomly draw 100, 000 samples from each of the 20 trained stego SinGANs to prove the possibility of secret image leakage is less than 0.001%. But I think 20 is a small number, it is sufficient to prove the security.

---

> ### Author Response · Authors · 2022-07-29
> **Response to Reviewer ShzF**
>
> Thanks for recognizing the merits of our work and for helpful suggestions.
>
> **1. Regarding only $20$ images as the testbed:** The primary reason to use the $20$ images from the original SinGAN’s repository is that it is much easier to demonstrate the normal behaviour of the stego SinGAN in comparison to the original SinGAN for security analysis. Note that as a completely different hiding scheme, image-based steganographic analysis methods cannot be applied. After all, it is effortless to reimplement the proposed SinGAN-based image hiding scheme by minor modification of official SinGAN implementations [3][4]. And thus, the generality of the proposed method on a wide range of natural images can be easily verified. Moreover, in the domain of single-image generative models, it is common practice to demonstrate the model feasibility on only dozens of images due to the computational burden during training[1][2]. Similar to these works, it is sufficient to train 20 images to show the effectiveness of our proposed solution.
>
> We are comfortable performing larger-scale experiments (for example, 200 image pairs) to further demonstrate the feasibility and effectiveness of the proposed hiding scheme, subject to the satisfaction of the reviewer.
>
>
> **2. Regarding one SinGAN for each cover/secret image pair:** We totally agree with the reviewer that the proposed method needs to train one SinGAN for each cover/secret image pair, which may be considered as a disadvantage. Nevertheless, the authors should point out that this is a general challenge in all secret-in-network data hiding works [5], and we are **the first** to propose an image-in-network hiding framework with improved extraction accuracy (see the third response on the exaggerated performance of HiNet), model security, and flexibility (hiding multiple images for different users, which is not accomplished before).
>
>
> **3. Regarding the exaggerated performance of HiNet and others in the current Table 1:**  The current HiNet obtains nearly perfect extraction accuracy in terms of PSNR ($\ge 45$ dB) and SSIM ($\ge 0.99$) in the original paper and in Table 1 of our manuscript. After careful re-examination of the official HiNet implementations and personal communication of the original authors, we find that the stego image by HiNet is not quantized to $8\times 3$ bpp (three for RGB channels) for transmission. Instead, each pixel of the stego image is the single-precision floating-point format of $32\times 3$ bpp. This accidentally incorrect implementation allows for a trivial hiding solution: we have more space to accommodate the cover and secret images by simple concatenation. The results of HiNet after quantization is shown in the table below.
>
> | Method   | PSNR  | SSIM  | DISTS |
> |----------|-------|-------|-------|
> | Baluja17 | 23.75 | 0.853 | 0.109 |
> | HiDDeN   | 25.89 | 0.875 | 0.106 |
> | Weng19   | 33.98 | 0.935 | 0.057 |
> | HiNet    | 32.53 | 0.935 | 0.054 |
> | Ours     | 34.58 | 0.951 | 0.039 |
>
>
> **4. Regarding low PSNR when obfuscating the image:** Thanks for pointing it out. We have compared our method with HiNet (with proper quantization) in the presence of image obfuscation. The results are listed in the below table, which we find that our method significantly outperforms HiNet.
>
> | Method | PSNR  | SSIM  | DISTS |
> |--------|-------|-------|-------|
> | HiNet  | 16.43 | 0.398 | 0.254 |
> | Ours   | 20.68 | 0.722 | 0.179 |
>
> **5. Regarding one-to-one mapping between the secret noise and the secret image:** We agree with the reviewer that it is difficult to mathematically ensure the bijective mapping between the secret noise and the secret image. Thus, we designed the experiment to empirically study the possibility of secret image leakage. As suggested by the reviewer, we will test this aspect of model security using a much larger image pairs, and update the results accordingly.
>
>
> **6. Regarding Line 97 and Lines 197-201:**
> The proposed method can be treated as both secret-in-network hiding, where the secret is a natural image, and constructive image hiding, in the sense that we hide a secret image during the construction  of a probability density function. In terms of extraction accuracy, we have updated results with corrected implementations of Baluja17, HiDDeN, Weng19, and HiNet, and modified the descriptions accordingly.
>
> [1] Shaham et al., SinGAN: Learning a generative model from a single natural image. In IEEE/CVF International Conference on Computer Vision, pages 4570–4580, 2019.
>
> [2] Hinz et al., Improved techniques for training single-image GANs. In IEEE/CVF Winter Conference on Applications of Computer Vision, pages 1300–1309, 2021.
>
> [3] https://github.com/tamarott/SinGAN
>
> [4] https://github.com/tohinz/ConSinGAN
>
> [5] Uchida et al., Embedding watermarks into deep neural networks. In ACM on International Conference on Multimedia Retrieval, pages 269–277, 2017.

---

### Official Review · Reviewer_4fNE · 2022-07-22

**Rating:** 3
**Confidence:** 4
**Soundness:** 2 fair
**Presentation:** 2 fair
**Contribution:** 1 poor

**Summary:**

This paper describes a technique for hiding images in images by means of SinGAN. The authors learn how to create a key-generated noise to be embedded into the cover image by exploiting SinGAN. Also discussion on robustness of the technique on discoverability is presented with some hints on obfuscation.

**Questions:**

1) Why did you use only SinGAN images for cover/secret? Can you do/propose some more examples?
2) How you carried out comparison with SOTA techniques? Did you do experiments on the same images or just reported each SOTA techniques paper results (I found the same PSNR/SSIM values in HiNet paper... I dont feel like this is a fair comparison).


**Limitations:**

The authors do not take adequately care of image size/resolution. In my opinion this is a great limitation of the work that should be addressed. Images elaborated are too small/low resolution to expose visible alterations... Maybe this is a limit... But we dont know because further experiments need to be carried out.

**Strengths And Weaknesses:**

The main strength of this paper is the application of the SinGAN architecture to the problem of image hiding. This is a first.
There are several weaknesses on the other hand:
1) the paper looks like this is a preliminary result, they employed only the 20 images available in the SinGAN Github repository;
2) some part of the text are not clear: (i) the authors claim to put the secret into a "location" of the cover... where? how? not clear to me... (ii) experiments were carried out on the 20 images of the SinGAN repository but what about the comparing SOTA techniques? They are comparable? I know that HiNet works with images with higher dimensions and quality... I dont feel like the comparison is fair
3) experiments need to cover much more possibilities
4) usual stego metrics are not shown like "db" or "bpp"
5) Figure 2 is shown but not reference in text

---

> ### Author Response · Authors · 2022-07-29
> **Response to Reviewer 4fNE  (Part 1)**
>
> Thanks for spending time and effort in providing the comments, which the authors highly appreciated. However, the reviewer seems to misunderstand the key contribution of our paper and we humbly disagree with the reviewers’ claim. We would like to clarify our key contributions and novelties as follows.
>
> **1. Regarding Novelty:** As well recognized by Reviewer bS3X, the primary contribution of the paper is a novel general computational framework for probabilistic image hiding (also can be considered as a form of image-in-network hiding and constructive hiding), which significantly departs from existing autoencoder-based image-in-image hiding schemes. In principle, the proposed framework can be implemented by a variety of deep probabilistic models, including diffusion-based and autoregressive models, provided that the guided sampling (illustrated in Fig. 1 (d)) can be feasibly designed. The proposed SinGAN approach is just a working instantiation of the more general hiding framework. Our framework has the advantages over the autoencoder-based hiding scheme in four ways. First, there is no need to communicate privately with the receiver the decoding network, which may be substantially large than the images to be hidden. Second, the proposed hiding framework is more secure because 1) it naively bypasses existing image-based steganographic analysis tools, traditional or deep, and 2) it demonstrates normal SinGAN behaviours in various ways. Third, it has improved extraction accuracy than autoencoder schemes, e.g., Baluja17, HiDDeN, Weng19, and HiNet. Note that after re-examining the current implementations of autoencoder schemes and personal communication of some of the original authors, we find all of them do not quantize the stego images into 24 bpp, leading to trivial hiding solutions and exaggerated performance (see the detailed analysis below). Fourth, our framework is capable of hiding multiple images for different receivers, a very challenging task that has not been accomplished before.
>
> **2. Regarding choosing the images from the SinGAN’s repository as the testbed:** We respectfully and firmly disagree the comment that `` the paper looks like this is a preliminary result’’. The primary reason to stick to the SinGAN’s repository is that it is much easier to demonstrate the normal behaviours of the stego SinGAN in comparison to the original SinGAN for model security analysis. Note that as a completely different hiding scheme, image-based steganographic analysis methods cannot be directly applied. After all, it is effortless to reimplement the proposed SinGAN-based image hiding scheme by minor modification of official SinGAN implementations [3][4]. And thus, the generality of the proposed method on a wide range of natural images can be easily verified. Moreover, in the domain of single-image generative models, it is common practice to demonstrate the model feasibility on only dozens of images due to the computational burden during training [1][2]. Similar to these stuides, it is sufficient to train 20 images to show the effectiveness of our proposed solution. We are comfortable performing larger-scale experiments (for example, 200 image pairs, which amounts to train 200 SinGAN generative models) to further demonstrate the feasibility and effectiveness of the proposed hiding scheme, subject to the satisfaction of the reviewer.
>
> **3. Regarding the location to hide the secret image in deep probabilistic models:** In the SinGAN instantiation, as we are working with implicit generative models, the location is implicitly determined by the fixed set of noise maps (fully determined by the embedding key) as inputs to the SinGAN. The way we hide the secret image is to fit a deterministic mapping from the fixed set of noise maps to the secret image during patch distribution learning. This corresponds to minimize the second reconstruction term in Eq. (4).
>
> [1] Tamar Rott Shaham, Tali Dekel, and Tomer Michaeli. SinGAN: Learning a generative model from a single natural image. In IEEE/CVF International Conference on Computer Vision, pages 4570–4580, 2019.
>
> [2] Tobias Hinz, Matthew Fisher, Oliver Wang, and Stefan Wermter. Improved techniques for training single-image GANs. In IEEE/CVF Winter Conference on Applications of Computer Vision, pages 1300–1309, 2021.
>
> [3] https://github.com/tamarott/SinGAN
>
> [4] https://github.com/tohinz/ConSinGAN

---

> > ### Author Response · Authors · 2022-07-29
> > **Response to Reviewer 4fNE (Part 2)**
> >
> > **4. Regarding the exaggerated performance of HiNet and others in the current Table 1:** The current HiNet obtains nearly perfect extraction accuracy in terms of PSNR ($\ge 45$ dB) and SSIM ($\ge 0.99$) in the original paper and in Table 1 of our manuscript. After careful re-examination of the official HiNet implementations and personal communication of the original authors, we find that the stego image by HiNet is not quantized to $8\times 3$ bpp (three for RGB channels) for transmission. Instead, each pixel of the stego image is the single-precision floating-point format of $32\times 3$ bpp. This accidentally incorrect implementation allows for a trivial hiding solution: we have more space to accommodate the cover and secret images by simple concatenation. The results of HiNet after quantization is shown in the table below.
> >
> > | Method   | PSNR  | SSIM  | DISTS |
> > |----------|-------|-------|-------|
> > | Baluja17 | 23.75 | 0.853 | 0.109 |
> > | HiDDeN   | 25.89 | 0.875 | 0.106 |
> > | Weng19   | 33.98 | 0.935 | 0.057 |
> > | HiNet    | 32.53 | 0.935 | 0.054 |
> > | Ours     | 34.58 | 0.951 | 0.039 |
> >
> >
> > **5. Regarding fair comparison to existing methods:** Thanks for bringing up the issue of comparing the proposed framework to existing ones. As also realized by Reviewer bS3X, the proposed probabilistic hiding framework and its SinGAN instantiation are the first of their kind, which are completely different from previous schemes. As a result, the proposed method cannot be directly compared to previous schemes, let alone fair comparison. As well recognized and appreciated by Reviewer bS3X, the authors try their best to identify the closest baselines (i.e., Baluja17, HiDDeN, Weng19, and HiNet), and test them on the same 20 image pairs (as shown in the Appendix) for extraction accuracy. The proposed method outperforms them in terms of extraction accuracy (see the above corrected and updated table) and efficiency. Moreover, the authors proposed three different computational tests to evaluate the model security.
> >
> >
> > **6. Regarding more possibilities in experiments:** We have covered a number of possibilities, and we are glad to cover more possibilities in our future work.
> >
> > **7. Regarding stego metrics like "db" and "bpp":** As a completely different hiding framework, no stego image (i.e., cover + secret image) is generated, and thus no stego metrics such as "db" and "bpp" can be computed. Instead, we only have the stego SinGAN to be publicly transmitted, whose embedding capacity cannot be directly measured by "bpp". To the best knowledge of the authors, there is no stego metric for image-in-network hiding, and it is of interest define one in the future, as suggested by the reviewer.
> >
> > **8. Regarding input image size/resolution:** One significant advantage of SinGAN is that it can work with and generate images of arbitrary resolution. The output resolution can even be different from the input resolution (e.g., using a $256\times 256$ training image to generate $1,024\times1,024$ images). The proposed image hiding scheme directly inherits the advantage from SinGAN, and thus the image size/resolution is not an issue. Nevertheless,
> > we agree with the reviewer that different image sizes/resolutions should be taken care of, and we will provide experimental results for images with high resolution (e.g., $1,024\times1,024$) in the Appendix.

---

### Meta-Review · Area_Chair_uG6H · 2022-08-30

**Recommendation:** Accept
**Confidence:** Certain

**Metareview:**

This paper studies a novel variation of image steganography. The proposed approach is different from prior work (mostly building on autoencoders) and uses a GAN and hide a secret image in one particular location of the learned distribution.

The central idea of the paper seems novel and interesting. The reviewers raised several concerns about limited evaluation and complexities of comparing to other methods that generate directly images. Overall, this paper seems to have novelty and interesting ideas and the benefits seem to overcome the limitations, based on the rebuttal and discussions.


**Award:**

No

---

### Decision · Program_Chairs · 2022-09-14

Accept